# Youth Participation for Sustainable Value Creation: The Role and Prioritization of SDGs

Tatjana Borojević [1], Nataša Petrović [2], Jelena Andreja Radaković [2,*], Hajdana Glomazić [3], Milan Radojičić [2], Nemanja Milenković [2], Damjan Maletič [1] and Matjaž Maletič [1]

1 Faculty of Organizational Sciences, University of Maribor, 4000 Kranj, Slovenia; tatjana.borojevic@student.um.si (T.B.); damjan.maletic@um.si (D.M.); matjaz.maletic@um.si (M.M.)
2 Faculty of Organizational Sciences, University of Belgrade, 11000 Belgrade, Serbia; natasa.petrovic@fon.bg.ac.rs (N.P.); milan.radojicic@fon.bg.ac.rs (M.R.); nemanja.milenkovic@fon.bg.ac.rs (N.M.)
3 Institute of Criminological and Sociological Research, 11000 Belgrade, Serbia; hajdana.glomazic@iksi.ac.rs
* Correspondence: jelenaandreja.radakovic@fon.bg.ac.rs

**Abstract:** Young people play a pivotal role in instigating and driving significant changes. The impact of individuals' involvement in environmental conservation, community development, and social change initiatives for sustainable value creation is of considerable importance in the pursuit of the sustainable development goals (SDGs). The objective of this study is to examine the contribution of youth engagement in the creation of sustainable value by analyzing their comprehension and prioritization of the 17 SDGs, with the goal of achieving sustainable development and sustainability. The researchers conducted a study on a sample of 1085 individuals between the ages of 14 and 30, who were selected from three distinct countries within the Western Balkan region—the Republic of Serbia, Bosnia and Herzegovina, and the Republic of Northern Macedonia. The data were analyzed using factor analysis in conjunction with descriptive and inferential statistics. The survey findings indicate that poverty, hunger, and access to adequate healthcare are the primary SDGs and societal challenges confronting the youth in these nations. Furthermore, it emphasizes the significance of both formal and informal education among youth as a catalyst for societal transformation for sustainable value creation.

**Keywords:** youth; youth participation; sustainable value creation; sustainable development goals; prioritization of sustainable development goals





## 1. Introduction

> *"Much did I rage when young,*
> *Being by the World oppressed,*
> *But now with flattering tongue*
> *It speeds the parting guest."*
> —W.B. Yeats, "Youth and Age".

Contemporary civilization, as well as local communities and people, is consistently faced with a multitude of challenges, with a significant portion of these challenges being of an environmental nature. Furthermore, there are other paradoxes, risks, and obstacles that remain unidentified, yet they are intricately linked to the prospective trajectory of humanity and the familiar global landscape. Martin, Maris, and Simberloff [1] argue that the mitigation of global risks necessitates the prioritization of environmental conservation and sustainability. Petrovic et al. [2] emphasize the significance of natural resources in facilitating sustainable economic growth and their inherent worth to humankind.

The subject of how to implement and maintain changes necessary for achieving the sustainable development goals (SDGs) (In continuation of the Millennium Development Goals (MDGs), the SDGs seek to foster worldwide mobilization in order to execute the

United Nations' 2030 Agenda for Sustainable Development. In the year 2015, all member nations of the United Nations (UN) made a collective commitment to the 2030 Agenda, therefore endorsing the 17 SDGs [3].)) is a crucial question within the realm of sustainability management. This question has significant importance from both a research and practical standpoint [4].

Despite the existence of several methodological approaches and prior research on the SDGs conducted by Komiyama and Takeuchi [5], academics continue to face challenges in identifying novel approaches to explain in detail the process of transitioning towards the SDGs, particularly when considering the viewpoint of the youth generation.

Considering the aforementioned factors and the heightened attention and regard for the Earth, the discipline of sustainability science emerged as a distinct area of study in the early 21st century [4–7]. This field encompasses and integrates various disciplinary, interdisciplinary, and transdisciplinary elements [8]. In recent years, there has been significant progress in the field of sustainability, as evidenced by the extensive research conducted by various scholars [5,9–11]. This progress has been achieved through the integration of multiple scientific disciplines, including ecology, biology, sociology, psychology, demography, technological studies, and history [11].

Sustainable development (SD) "as a concept was developed alongside acute awareness that ecological destruction and the 1980s' retreat from social concerns'—manifested as poverty, deprivation, and urban dereliction that blight many parts of the world—are untenable" [12]. SD is an idea that has emerged over the past few decades and has been articulated most notably in key documents such as the 1972 Stockholm Declaration, the 1987 report "Our Common Future" by the World Commission on Environment and Development of the United Nations, the 2001 Earth Summit in Rio de Janeiro, and the third UN Conference on Environment and Development held in Johannesburg in 2002. The papers emphasize the pivotal role of young people and advocate their active involvement in driving this growth. The youth have been recognized as a crucial component in the existing SD framework and the actualization of the SDGs. The terms "children", "young", and "youth" are referenced in a total of 33 occasions within the SDGs. Notably, at least 10 out of the 17 SDGs exhibit a clear correlation with this demographic group, encompassing their growth and advancement [13]. Additionally, a Youth Speak global survey from 2016 that encompassed around 180,000 young individuals coming from 126 different countries showed that 68% of them believed that the world will be a better place by the year 2030, showing their driving force and optimism to change the world for the better and fix the mistakes of the past [14].

Furthermore, it has been acknowledged that young people are seen as both social and moral actors and are accepted as fully fledged democratic members of society [15].

The significance of the involvement of young people in achieving all the SDGs is specifically highlighted in Agenda 21 [16]. This Agenda examines the engagement of many stakeholders in critical sustainability processes, including openness, transparency, and democracy [11–15]. Additionally, it emphasizes the significance of young people as both the current and future members of our society. They are seen as a valuable source of innovation and a driving force behind progress in society. Hence, it is imperative to consistently and methodically allocate resources towards the advancement of youth development and foster a collaborative relationship between youth and the state. This approach aims to augment the active engagement of young individuals within society, promote their social assimilation, and guarantee their involvement in the formulation of youth-oriented policies.

Having said that, there is still not enough research on SDG prioritization, especially prioritization demonstrated by youth as an age category, or comparative nations, and this is precisely why the authors of this paper chose this subject—to expand the body of knowledge on the subject of youth prioritization of SDGs and to see how comparative nations of the Western Balkans view the importance of different SDGs.

The participation of the youth, who represent the next generation of ecological, economic, and social decision-makers, is essential in achieving all of the SDGs set forth in 2015

by the United Nations. This is primarily because, as authors Raikes and others noted in 2017 [17], they will experience the greatest effects from future advancements made in all areas of the SDGs. In this context, it is imperative to point out that just providing young people with information regarding sustainability, SD, and its objectives is inadequate. Instead, it is important to actively involve and immerse them in all the important processes of efficient development. This will help us deal with the many problems we are currently facing with SD. The rationale for this assertion stems from the observation that it is the younger generation who bears the responsibility of establishing a connection between efficacious sustainability objectives and the requisite measures to achieve them [18].

Also, getting young people involved in community activities that help reach and use the SDGs, SD, and sustainable value creation has positive effects not only for the community but also for the young people involved. The present literature lacks a broadly recognized definition of sustainable value generation, highlighting the need for further clarification in this area. Various methodologies concur that the concept in question is a complete one since it covers value in all facets of SD. Consequently, it considers the concerns and preferences of many stakeholders, e.g., [19–21]. In other words, "in this vein, it can be understood as organizational contributions to achieving the SDGs" [21].

This involves the active participation of young people in community matters (such as environmental sustainability) and is recognized as one of several processes that support positive youth development [22–24].

In addition, it must be said that when it comes to the SDGs and their achievement, it is necessary, bearing in mind that there are 17 of them with 169 sub-goals, to assign priorities to each of them at the national, local, and individual levels. "In this context, it is important to note that the capacity to fully implement all SDGs may not be available, which means that prioritization may become a necessary component of the policy process" [25,26]. This prioritization is "quite realistic" because it corresponds to the current state of the environment, society, and economy, and enables efficient and effective activities to achieve the goals; in this case, for the selected ones, prioritization, i.e., ranking, was performed. Bearing in mind the very significant role that young people play in the implementation of the SDGs and their realization, a role is logically imposed.

There exists a restricted body of knowledge that thoroughly examines the strategies for boosting adolescent engagement in the successful execution and attainment of SDGs, specifically within the Southeast European territory. Also, there is a limited body of research on value creation for sustainability, as the existing research mostly focuses on the idea of social value creation, e.g., [27,28], as well as a scarcity of literature related to the participation of youth in this particular domain.

The limited availability of studies has been the driving force for the selection of this study topic. Our research aims to address the existing knowledge gap in the literature pertaining to this topic. It is essential to note that this study represents an early exploration of the significance of prioritizing SDGs in fostering young participation in the pursuit of sustainable value creation.

The subsequent sections of the paper are structured in the following manner: Following the introductory section, a literature review and a theoretical background of the research is presented. After that, the paper proceeds to provide a detailed account of the materials and methods used in the study. The next section of the research relates to the presentation and analysis of the findings derived from our research. The final section of the study includes the discussion and conclusion, and practical implications and possible directions for further research are presented.

## 2. Theoretical Background

### 2.1. Youth Participation

In today's contemporary society, the youth face numerous challenges in their inclusion in the social and political life. Challenges such as unemployment, discrimination, poverty, and social exclusion often prevent young people from actively joining social life and

contributing to the community. They also encounter other obstacles, such as a lack of political education and information, an inability to access resources, a lack of support for their ideas and initiatives, and an absence of tailored activities that would enable them to join society in a manner appropriate for their age and interests. These challenges aggravate young people's access to social life, which can negatively affect their development and wellbeing.

Furthermore, the involvement of young individuals and their active participation in civic affairs serve as a means of jointly addressing current global difficulties [29], particularly those related to SD [30].

The topic of youth participation in the advancement of the SDGs has garnered significant attention in the recent academic literature [31]. Despite the importance of this issue, it can be claimed that there exist several unaddressed problems that need consideration to be relevant for current research in various contexts of youth participation. For example, within the framework of globalization, the active involvement of young people in societal affairs and their engagement in civic activities exert a serious influence on the functioning of democratic systems and the protection of democratic principles [32]. "The term 'youth' and 'young people' are used interchangeably in the SDGs to represent the voices of millennials towards sustainability" [33]. Also, the Lisboa+21 Declaration on Youth Policies and Programmes 2019 underscores the significance of young people and their contribution to the SDGs, keeping in mind their role in change as well as fact that the essences of SDGs are integration, indivisibility, and universality, "and therefore that all of them apply to youth" [34].

### 2.2. Prioritization of SDGs

Researchers have examined the prioritization of SDGs in the literature, keeping in mind that while the SDGs are fundamentally equivalent in importance to global and SD objectives, their implementation at the national and local levels depends on certain SDGs being prioritized.

The authors Forestier and Kim [35] claim that "efficient implementation depends largely on the good will of national governments", coinciding with the findings of the authors Biermann, Kanie, and Kim [36]. To this should be added the study of the authors Stevens and Kanie, which clearly indicates that the prioritization of SDGs "has enabled broad participation and support for the SDGs" [37]. What can certainly be concluded is that different governments add different priorities to certain SDGs, not only on the basis of national policies but also economic interests, as concluded by the authors Horn and Grugel [38]. Even the authors Linnerud and others claim that "global prioritization may not be entirely avoidable and in some cases even desirable when taking into account the differing contexts and capabilities of each country" [39].

Some researchers noted in their papers that the necessary reason for prioritization of SDGs in national practices lies in the fact that governments have tendencies for economically effective and quick achievement of SDGs, so they choose to prioritize them [40]. Many studies reached the conclusion that this approach in reaching SDGs at the national level comprises critical components, given that achieving certain SDGs tends to compromise or ignore others [35]. But even so it must be added that the prioritization of SDGs cannot be avoided because "insufficient capacity of many countries to fully implement all SDGs makes prioritization inevitable or even necessary" [35,41]. Furthermore, it is an undeniable fact that different governments prioritize particular goals over others [42].

In relation to the prioritization of SDGs by youths, it is important to note that while there have been several previous studies conducted on this subject, it is crucial to highlight the existing gap in the academic literature. This allows academics to explore further possibilities for research within this field of study. Furthermore, there is a lack of established methodologies for systematically studying the specific SDGs that are prioritized by different nations, local communities, and youth populations.

For example, the authors Chairattanawan and Somwang examined 125 students at Sripatum University in Bangkok, Thailand, and the students ranked the SDGs according to the following importance: SDG4 (quality education) was the first, the second was SDG13 (climate action), and the third was SDG3 (good health and wellbeing) [43]. The authors Borojević and others [44] had in their research a representative sample that included 1586 young people from the Republic of Serbia and they ranked the most important SDGs as follows: SDG2 (zero hunger), SDG3 (good health and wellbeing), and SDG5 (gender equality).

An interesting study was conducted by the authors Petković and others [45] on a research sample of 386 young people from the Republic of Serbia. Their study deals with forecasting the importance of SD pillars (economic, social, and environmental) using the ANFIS method, and the most important pillar ranked was environmental. A similar study and results were obtained by authors Gaur and others [46] with data that were collected from 425 youth respondents in India.

## 3. Materials and Methods

### 3.1. Sample Collection

To conduct this study, a representative sample of individuals aged 14 to 30 was selected using a random sampling approach in the municipalities of New Belgrade (the Republic of Serbia), Kumanovo (the Republic of North Macedonia), and Tuzla (Bosnia and Herzegovina).

The municipalities chosen for the purpose of this research represent the most densely populated regions in their respective countries, excluding the capital city. They have several common characteristics: firstly, they are all situated in the Western Balkans region; they possess a significant youth population that is characterized by ethnic, cultural, and religious diversity; and they actively participate in regional initiatives that seek to promote the achievement of SDGs. In addition, these countries share a collective objective of achieving membership in the European Union. Currently, they are actively engaged in the enactment and implementation of significant revisions to their individual local and national youth policies. As a result, the framework and legislation implemented by the respective governing bodies demonstrate a willingness to embrace innovative strategies in order to promote youth engagement. This, consequently, has the capacity to cultivate improved inclusiveness and involvement among young people. In contrast to the commonalities, these municipalities exhibit variations in terms of the range, structure, and scope of the local services they use and provide to attain certain SDGs.

The sample size of 1085 individuals was determined using an online sample size calculator [47] with a confidence level of 95% and a margin of error of 5%. The **data collection procedure** involved **an online questionnaire disseminated to** the **selected** sample, ensuring diversity in age groups.

The research was carried out only for scientific and scholarly objectives throughout the months of August and September 2023. The participants were provided with detailed information on the purpose and methodology of the research, as well as any relevant details required for them to make a well-informed and voluntary decision regarding their participation in the study. The individuals were provided with a guarantee of maintaining their confidentiality and informed consent was obtained from all subjects involved in the study.

### 3.2. Questionnaire

The questionnaire used in the study consisted of 117 questions grouped into two sections. The first section included seven questions focusing on demographic attributes such as gender, age, education, marital and parental status, employment, and place of residence. The second section covered five areas, awareness, knowledge, attitude, and youth participation, where participants responded to statements using a 7-point Likert scale. The last question was related to ranking the 17 SDGs based on their perceived relevance.

In order to test the reliability and internal consistency of the measurement scale, a reliability analysis was conducted using Cronbach's alpha coefficient. The analysis was performed both on individual groups of questions and on all questions collectively. The results indicate an excellent level of reliability for the measurement scales with values exceeding 0.9 [48].

*3.3. Sample Composition*

The research project encompassed a sample size of 1085 individuals, providing a representation of the youth population across three municipalities situated in three distinct countries. The sample was constructed using the most recent census data from each country or municipality, with careful consideration given to ensure that the sample accurately represented the population based on the demographic features of the respondents. The data for the sample were derived from the reference data provided by the statistical offices of each country/municipality [49–51].

The sample size is given in Table 1.

**Table 1.** The sample size.

| | Municipality/Country | | |
|---|---|---|---|
| | **Novi Beograd/ The Republic of Serbia** | **Kumanovo/ The Republic of North Macedonia** | **Tuzla/ Bosnia and Herzegovina** |
| Total number of young people in municipality | 37,520 | 30,057 | 22,134 |
| The sample size of the youth population that participated in the research | 366 | 359 | 360 |
| Percentage of sample size in total youth population of the municipality | 0.97 | 1.19 | 1.63 |
| Percentage of country's youth population in total country population | 15.76 | 23.56 | 21.78 |

A concise presentation of the demographic characteristics of the sample is as follows:

- The research sample, comprising 1085 individuals, reflects a well-balanced distribution across the three countries: 33.7% from the Republic of Serbia (366 respondents), 33.2% from Bosnia and Herzegovina (360 respondents), and 33.1% from the Republic of North Macedonia (359 respondents). A Chi Square Test ($\chi^2$ = 0.079, df = 2, $p$ = 0.961) confirms the uniformity in the country-of-origin representation.
- In terms of gender, the sample demonstrates equality, with 49.8% identifying as male (540 participants) and 50.2% as female (545 participants). The gender distribution is statistically uniform ($\chi^2$ = 0.023, df = 1, $p$ = 0.879).
- The age distribution of respondents spans from 15 to 30 years, with 30.7% falling in the 15–19 age category (333 respondents), 38.2% in the 20–24 age category (415 respondents), and 31.1% between 25 and 30 years old (337 respondents).
- Regarding educational attainment, 2.0% of participants (22 individuals) completed primary school, while the majority, 73.1% (793 individuals), finished secondary school. Additionally, 24.1% (261 individuals) achieved a higher education level, and 0.8% hold a master's or doctoral degree (9 respondents).
- Participants' marital status varies, with 71.1% identifying as unmarried (771 respondents), 17.0% as married (184 respondents), and 12.0% reporting having a partner.
- Concerning parental status, 77.5% of respondents (841 individuals) do not have children. Among those with children (22.5%), 13.1% reported having one child, while 9.4% have two or more children.
- In terms of employment status, the sample consists of 60.2% unemployed individuals (653 respondents) and 39.8% employed individuals (432 respondents).

- The residential distribution indicates that 73.7% of respondents (800 individuals) reside in urban regions, while 26.3% (285 individuals) live in rural areas.

### 3.4. Statistical Analysis

The statistical program IBM SPSS ver. 23 was utilized for data processing and analysis involving descriptive and inferential methods. The questionnaire's reliability was assessed using Cronbach's alpha coefficient. The assessment of the questionnaire's dimensionality was conducted using factor analysis. Relying on the central limit theorem, we applied one-way ANOVA to examine differences between countries, while post hoc comparisons were conducted using the Tukey post hoc test. The Chi Square Test was utilized to test the uniformity of the sample composition through a comparison of theoretical frequencies with empirical frequencies.

## 4. Results

Factor analysis is a valuable statistical technique that may be employed to examine the interconnectivity of the SDGs. The SDGs exhibit interconnections and frequently share aims, hence enabling the use of factor analysis as a method to discover underlying com-ponents or themes that may signify clusters of interconnected goals.

In order to reduce the dimension of the observed problem and see how the questions are grouped with each other according to importance, the dimensionality of the section related to the prioritization of SDGs is reduced by applying factor analysis with Direct Oblimin rotation. By employing the Kaiser–Meyer–Olkin measure of sampling and Bartlett's Test, we concluded that the dataset is suitable for factor analysis. KMO exceeds the threshold value of 0.6, equaling 0.882, and the value of Bartlett's Test of Sphericity is statistically significant ($p < 0.001$). To determine the number of factors, we used parallel analysis [52]. Four factors were retained, which together explain 58.09% of the variance. Looking at them separately, the first factor explains 28.37%, the second 13.48%, the third 9.26%, and the fourth explains 6.97% of the variance. Table 2 shows factor loadings on four retained factors.

**Table 2.** Factor loadings.

| Item Number | Item/SDGs | Component | | | |
|---|---|---|---|---|---|
| | | 1 | 2 | 3 | 4 |
| p3 | Good health and wellbeing | 0.984 | | | |
| p2 | Zero hunger | 0.978 | | | |
| p1 | No poverty | 0.893 | | | |
| p13 | Climate action | −0.417 | | −0.343 | |
| p12 | Responsible consumption and production | | 0.638 | | |
| p9 | Industry, innovation and infrastructure | | 0.637 | | |
| p11 | Sustainable cities and communities | | 0.621 | | |
| p16 | Peace, justice and strong institutions | −0.396 | −0.597 | | |
| p7 | Affordable and clean energy | | | 0.684 | |
| p10 | Reduced inequalities | | | −0.629 | |
| p8 | Decent work and economic growth | | 0.424 | 0.517 | |
| p6 | Clean water and sanitation | | | 0.502 | |
| p4 | Quality education | | | | 0.779 |
| p5 | Gender equality | | | | 0.759 |
| p14 | Life below water | | | | 0.694 |
| p15 | Life on land | | −0.380 | | 0.579 |
| p17 | Partnership for the goals | | | | 0.487 |

The first factor is composed of four elements: *(1) Good health and wellbeing, (2) Zero hunger, (3) No poverty,* and *(4) Climate action.* Factor loadings for the first factor range from −0.417 to 0.984. This factor is titled global sustainable goals.

The second factor has factor loadings from −0.597 to 0.638. It is made up of the following areas: *(5) Responsible consumption and production, (6) Industry, innovation and infrastructure, (7) Sustainable cities and communities,* and *(8) Peace, justice and strong institutions*, and, given its contents, it is called responsible building of society.

The third factor is made up of the following aspects: *(9) Affordable and clean energy, (10) Reduced inequalities, (11) Decent work and economic growth,* and *(12) Clean water and sanitation.* They have factor loadings from 0.502 to 0.684. This third factor is called resource availability.

The fourth factor, titled sustainable inclusion and living environment, is composed of the following items: *(13) Quality education, (14) Gender equality, (15) Life below water, (16) life on land*, and *(17) Partnership for the goals,* with factor loadings from 0.487 to 0.779.

The questions regarding youth participation are shown in Table 3. Respondents were asked to express their agreement with each individual item on a 7-point Likert scale (1 = totally disagree; 7 = absolutely agree). A higher total score on the four isolated dimensions of the questionnaire indicates a stronger agreement with the content within those dimensions.

**Table 3.** Reliability and descriptive indicators of items in the questionnaire: youth participation in the achievement of SDGs.

| | N | Min | Max | M | SD | Cronbach's Alpha |
|---|---|---|---|---|---|---|
| The present generation should ensure that the next generation can live in communities that are at least equally healthy as the present ones | 1085 | 1 | 7 | 5.82 | 1.24 | 0.92 |
| Donating money for certain areas and SDGs is an efficient form of support | 1085 | 1 | 7 | 5.70 | 1.27 | 0.90 |
| Volunteering to support certain areas and SDGs yields efficient results | 1085 | 1 | 7 | 5.67 | 1.27 | 0.90 |
| Participation in benefit concerts in support of some areas and SDGs is a good example of support | 1085 | 1 | 7 | 5.67 | 1.27 | 0.90 |
| By taking part in the work of different institutional bodies on the local level I can impact change in different areas and SDGs | 1085 | 1 | 7 | 5.66 | 1.28 | 0.90 |
| Taking part in petitions in support of some areas and SDGs enables change | 1085 | 1 | 7 | 5.65 | 1.32 | 0.90 |
| Boycotting or purchasing certain products to support SD and its goals is an efficient model of support | 1085 | 1 | 7 | 5.49 | 1.37 | 0.91 |

Respondents show a high degree of agreement with the first dimension, global sustainable goals: 5.47 (std = 0.98). Within this dimension, they agree the most with the item *No poverty*, 6.22 (std = 1.54), and the least with *Climate action*, 3.36 (std = 1.58).

Responsible building of society as the second factor has the medium average grade of 3.59 (std = 0.76), i.e., it is ranked in the middle of the prioritization pyramid. *Industry, innovations and infrastructure* is the item within this factor with the highest degree of agreement, 4.29 (std = 1.42).

The third factor, Resource availability, is important in the prioritization of SDGs, 4.80 (std = 0.65), while *Clean water and sanitation* is the most important priority within this dimension, 5.51 (std = 1.24).

Sustainable inclusion and living environment, as the fourth factor, is of medium importance, 3.67 (STD = 0.58), to youth from the Republic of Serbia, Bosnia and Herzegovina,

and the Republic of North Macedonia. *Quality education* is the item respondents find the most important within the fourth dimension (5.62 (std = 1.35)).

In summary, youth prioritize the most the factor Global sustainable goals (5.47 (std = 0.98)), followed by Resource availability (4.80 (std = 0.65)) and Sustainable inclusion (3.67 (std = 0.58)), while Responsible building of society is seen as the least important (3.59 (std = 0.76)).

The next analysis employed was one-way ANOVA to investigate potential differences in awareness, knowledge, and attitudes toward SD among youth across three countries. The analysis aimed to assess whether there were statistically significant variations in these dimensions among the youth populations of the Republic of Serbia, Bosnia and Herzegovina, and the Republic of North Macedonia. The results are shown in Table 4.

**Table 4.** Awareness, knowledge, and attitudes toward SD per country (ANOVA).

| Variable | Mean | std | F | *p* |
|---|---|---|---|---|
| *Awareness* | | | 119.55 | 0.000 |
| Republic of Serbia | 6.14 | 0.89 | | |
| Bosnia and Herzegovina | 5.83 | 0.77 | | |
| Republic of North Macedonia | 5.18 | 0.88 | | |
| *Knowledge* | | | 138.28 | 0.000 |
| Republic of Serbia | 6.18 | 0.86 | | |
| Bosnia and Herzegovina | 5.86 | 0.85 | | |
| Republic of North Macedonia | 5.13 | 0.91 | | |
| *Attitudes* | | | 117.57 | 0.000 |
| Republic of Serbia | 6.16 | 0.96 | | |
| Bosnia and Herzegovina | 5.90 | 0.88 | | |
| Republic of North Macedonia | 5.16 | 0.90 | | |

The results indicate a statistically significant difference in awareness, knowledge, and attitudes toward SD among the youth populations of the countries in question, as determined via ANOVA. Further analysis using the Tukey post hoc test revealed significant differences between every country across these dimensions. Youths of the Republic of Serbia have the highest mean score in every segment. The youths from the Republic of North Macedonia have the lowest mean score. (The potential factors contributing to greater SDG awareness in the Republic of Serbia and Bosnia and Herzegovina, in comparison to the Republic of North Macedonia, might be the following: (1) the allocation of resources for the promotion of SDGs in the first two countries might be more substantial and well established in comparison to the Republic of North Macedonia, (2) the role of media and communication in shaping public awareness varies among countries due to differences in media coverage, encompassing online platforms, traditional media outlets, and social media initiatives, and (3) the disparities in the effectiveness and amount of participation of civil society organizations and community-level efforts in advancing the SDGs could potentially influence the levels of awareness achieved.)

The main part of the research was finding out how the participants from the younger demographic assign a subjective ranking to the 17 SDGs based on their perceived relevance to both the person and the local society. The ranking system utilized a scale wherein a rank of 1 denoted the highest level of importance, while a rank of 17 indicated the lowest level of importance for a certain sustainable development goal (SDG). The results of their rankings are presented in Table 5.

**Table 5.** Youth prioritization of 17 SDGs.

| Rank | SDG | Mean |
|---|---|---|
| 1 | SDG1 No poverty | 3.84 |
| 2 | SDG2 Zero hunger | 4.08 |
| 2 | SDG3 Good health and wellbeing | 4.08 |
| 4 | SDG4 Quality education | 5.55 |
| 5 | SDG6 Clean water and sanitation | 6.48 |
| 6 | SDG5 Gender equality | 6.91 |
| 7 | SDG7 Affordable and clean energy | 7.93 |
| 8 | SDG8 Decent work and economic growth | 8.57 |
| 9 | SDG10 Reduced inequalities | 9.63 |
| 10 | SDG9 Industry, innovation, and infrastructure | 9.70 |
| 11 | SDG11 Sustainable cities and communities | 11.05 |
| 12 | SDG13 Climate action | 11.61 |
| 13 | SDG12 Responsible consumption and production | 11.82 |
| 14 | SDG16 Peace, justice, and strong institutions | 11.87 |
| 15 | SDG15 Life on land | 12.34 |
| 16 | SDG17 Partnership for the goals | 13.71 |
| 17 | SDG14 Life bellow water | 13.83 |

## 5. Discussion

The process of ranking the SDGs has the potential to enhance the participation of young individuals in the world's pursuit of SD. By prioritizing certain goals, this method can effectively engage young people, directing their attention and inspiring them to take action. By means of education, dialogue, study, and action, young individuals have the potential to actively participate in the ranking process, thereby influencing their own values and priorities. The effective prioritization of SDGs, which encompasses the ability of young individuals to assess and evaluate the goals based on their significance, is influenced by various aspects, such as the following:

- Access to the education system, both official and informal, plays a crucial role in the development of capabilities necessary for prioritizing the SDGs. Formal education offers a fundamental comprehension of worldwide difficulties and objectives pertaining to SD, but informal education, encompassing workshops, seminars, and educational campaigns, can foster a more profound cultivation of analytical skills and critical thinking among young individuals. Education facilitates individuals in comprehending the broader ramifications of diverse objectives and their interdependent influence, alongside the cultivation of values [53–55]. Individuals belonging to lower social and economic strata frequently encounter exacerbated barriers in terms of accessing education and acquiring the necessary resources essential for the cultivation of requisite participatory abilities. According to Finlay, Wray-Lake, and Flanagan [56], smaller and economically disadvantaged towns exhibit a dearth of infrastructure and restricted availability of youth-focused programming. The involvement of young individuals in social developments may be impeded by financial limitations and a dearth of Internet connectivity.

- The socioeconomic situation of individuals significantly influences the manner in which young individuals perceive and prioritize the SDGs. Individuals from diverse socioeconomic situations sometimes possess distinct viewpoints and prioritize varying

goals during their youth. Individuals with higher levels of socioeconomic resources tend to prioritize goals associated with economic development and prosperity. Conversely, individuals with a lower socioeconomic status tend to place a larger emphasis on goals pertaining to the reduction in poverty and inequality. Several studies have indicated that those with a higher socioeconomic position are more likely to participate in social interactions. Another study revealed that the higher the family income, the higher the score on the scale of ecological awareness and active participation [57]. Research studies have consistently demonstrated that parental socioeconomic position significantly influences the level of youth engagement and educational attainment [58]. Nevertheless, over the course of many decades, these elements have served as indicators for distinct patterns of behavior among young individuals. In previous decades, there was a strong correlation between a higher socioeconomic position and engagement in activism. However, this relationship diminished in the following decades. Given the current circumstances, it is imperative to reexamine this occurrence in order to obtain pertinent data.

- The influence of cultural values and norms is significant in defining the priorities of young individuals with regard to the SDGs. The patterns of youth engagement in society exhibit variations based on cultural norms, historical context, and geographical factors [59]. The attitudes of young individuals towards the goals frequently mirror the ideals inherent in their group and culture. In certain communities that prioritize nature preservation and traditional values, young individuals may exhibit a greater emphasis on the objectives of conserving the natural environment and safeguarding cultural heritage. Hence, the effective prioritizing of SDGs by young individuals is contingent upon their level of education, socioeconomic standing, and cultural norms within their respective communities. A comprehension of these elements facilitates the formulation of strategies and initiatives aimed at promoting the active engagement of young individuals in the pursuit of the SDGs.

As observed from the results, the youth ranked SDG1 End poverty in all its forms everywhere the highest. This may be due to the fact that the region of the Western Balkans, where the three countries in question are located, has high youth unemployment rates, and this SDG directly addresses their economic wellbeing. High youth unemployment rates cause an emigration of young individuals to foreign nations in pursuit of improved prospects [60]. Despite the observed economic development following the COVID-19 epidemic, this region continues to exhibit fragility due to the persistent issue of excessive unemployment. The exacerbation of this issue was compounded by the significant reduction in employment opportunities resulting from the pandemic, disproportionately impacting women and the younger demographic. Despite the implementation of many policies over a prolonged period, young individuals in the Balkans region persistently encounter challenges when attempting to access the labor market. This is evident from the prevailing rates of youth unemployment, which remain higher than the average rates observed throughout the European Union [61].

Consequently, the Western Balkan countries are confronted with significant emigration rates, accompanied by declining natality rates among the population. This demographic shift has resulted in a notable fall in the economic performance of these economies, and the younger generation has felt it first-hand [62,63].

Over the past few years, the Republic of Serbia's labor market has experienced significant changes regarding its indicators, as evidenced by a decline in the youth unemployment rate (15 to 24 years of age) from 34.9% in 2016 to 26.6% in 2020. However, young Serbian individuals face significant disadvantages when they enter the labor market, particularly those who are unemployed and not engaged in any form of education or training [61]. The findings by Alili, King, and Gëdeshi [64] as well as Vutsova, Arabadzhieva, and Angelova [65] pertaining to the Republic of North Macedonia and Bosnia and Herzegovina reveal that those who are more inclined to emigrate from the country are undergraduate students, as opposed to postgraduate students. Additionally, those who do not have inten-

tions to pursue higher studies and students with a familial background of migration are also more likely to depart the country looking for better labor opportunities.

SDG1 is closely ranked in importance to SDG2, which focuses on ending hunger, achieving food security, improving nutrition, and promoting sustainable agriculture, as well as SDG3, which aims to ensure healthy lifestyles and promote wellbeing for individuals of all ages. These last two SDGs share the rank of second priority. The issue of hunger is frequently interconnected with endeavors aimed at mitigating poverty. The younger generation may perceive the goal of achieving "Zero Hunger" as a potential strategy for mitigating poverty within their localities, fostering economic progress and societal cohesion.

On the other hand, ranking SDG3, which aims to ensure healthy lives and promote wellbeing for all at all ages, at second place does not come as a surprise according to a study from 2022 carried out by Maljichi and others [66]. The study's main findings indicate that there exists a diminished degree of trust among residents residing in the Western Balkans towards their healthcare system, and that most residents prefer and trust private care institutions far more than government healthcare. The youth period is commonly perceived as a stage of life characterized by good health, and it is true that a majority of individuals in this age group enjoy a state of wellbeing. Nevertheless, it is projected that more than 1.3 million individuals between the age range of 15 and 24 years die annually from preventable conditions [67,68] On the other hand, interpersonal violence ranks as one of the four most prevalent causes of death amongst individuals aged 15–29 years [69].

It is worth acknowledging that the high ranking of SDG6, which focuses on guaranteeing universal access to clean water and sanitation, can be attributed the fact that the region of the Western Balkans faces annual flooding events that result in significant damage to infrastructure and economic activities. Additionally, there has been a substantial rise in water pollution levels in the rivers of the region [70].

The recognition of the varying attitudes and priorities among young individuals is critical in addressing climate change, as it plays a crucial role in ensuring the sustainable future of our world. Despite the considerable importance of climate change as a worldwide concern, there exist various factors that may contribute to the relatively lower prioritization of the climate change SDG (SDG13) among certain young individuals. This could be due to the following:

- In areas where young individuals are directly impacted by urgent and immediate matters, such as poverty, unemployment, or societal conflicts, these issues may be prioritized over long-term global challenges, such as climate change. If individuals are confronted with more pressing issues, their attention may consequently be redirected.
- Young people in their formative years may not perceive the immediate ramifications of climate change within their proximate environment. In areas where the impacts of climate change are not yet as pronounced, people may allocate greater importance to other concerns that have a more immediate and tangible impact on their daily existence. This is slightly paradoxical, bearing in mind that Europe is facing never-seen-before heatwaves, with each year being hotter than the previous one [71]. This raises concerns about the standard of education in the area and whether or not the younger generation is aware of the correlation between rising temperatures and climate change.

According to the responses, the least important SDGs ranked were SDG15 Life on land, SDG17 Partnership for the goals, and SDG14 Life below water. The tangible effects of international cooperation and partnership may not be immediately apparent in the daily experiences of young individuals residing in the Western Balkans. The limited visibility associated with SDG17 could potentially lead to a diminished perception of its significance. This highlights the importance of increased youth participation in local governance. The active engagement of young individuals in addressing both local community and universal concerns is crucial, as it not only contributes to their personal development but also enhances the overall functioning and quality of any given society. According to Borojević and others [72], many young individuals fail to perceive themselves as being fundamentally

embedded within the societal framework in which they reside. They seem to disassociate from the notion that they are interconnected with the destiny of this system, and that they possess no direct agency over the events taking place in distant locations, contrary to their volition and aspirations.

On the other hand, the rationale behind the Western Balkan youth ranking SDG14 last is justified by the nature of this particular SDG, which pertains to the preservation and sustainable use of the oceans, seas, and marine resources for the purpose of SD. It is important to note that none of the three countries involved in the research possesses direct access to an ocean or sea, thereby explaining their lower ranking of this specific SDG.

## 6. Conclusions

Young people are the future of the planet. They are the future presidents, CEOs, innovators, and creators. It is imperative to provide comprehensive education to young individuals regarding all SDGs to enhance their comprehension of the fundamental principles and significance of these goals. This assertion has been previously acknowledged and emphasized by various scholars [73–75]. Unfortunately, recent research has indicated that the younger generation lacks adequate understanding pertaining to the SDGs. Consequently, it becomes imperative to explore potential strategies aimed at mitigating this difficulty. The data indicating that 58% of young research participants actively participate in environmental protection activities in their daily lives provide support for the argument that further education is necessary to facilitate the participation of young individuals in a manner that aligns with their personal preferences [27]. The prioritization of SDGs can serve as a useful strategy in achieving this objective, which will consequently enable sustainable value creation.

The prioritization of SDGs holds significant importance in fostering the participation and involvement of young individuals, owing to various compelling justifications:

- The SDGs encompass a total of 17 goals, and it is commonly observed that young individuals frequently encounter a surplus of information.
- The process of prioritizing the SDGs enables individuals to focus their attention on those that are most proximate and hold the greatest perceived impact.
- The act of young individuals prioritizing the SDGs would serve as a source of motivation and encouragement for them to actively participate in tangible actions.
- The probability of individuals actively participating and making efforts to accomplish their objectives is enhanced when they possess well-defined priorities.
- Through engaging in the process of ranking, young individuals may experience a heightened sense of participation in a broader collective endeavor, fostering a sense of affiliation and inclusion.
- The experience of perceiving alignment between one's personal priorities and those of their peers and community engenders a sense of belonging and support.
- The implementation of ranking systems will facilitate the comprehension of the primary objectives as perceived by young individuals, hence enabling organizations and governmental bodies to gain enhanced insights. This will facilitate a more efficient allocation of resources and the development of policies and activities that are in line with the needs of young individuals.

The study findings given in this paper have provided evidence that the engagement of young individuals in the advocacy and execution of the SDGs is characterized by a variety of dimensions and has significant effects. Young individuals contribute novel viewpoints, inventive concepts, and a distinctive dynamism to the discourse. They embody the roles of change-makers, advocates, and champions in the pursuit of sustainability. The involvement of individuals is not solely a question of fairness, but also a crucial strategic necessity for achieving SD.

Furthermore, the youth's prioritization of certain SDGs can be seen as an indication of their consciousness and dedication towards addressing the most urgent global issues. Young individuals are actively engaged in addressing various crucial goals, such as poverty

eradication, quality education, gender equality, climate change, and more, within their communities, regions, and beyond. The selection of their priorities serves as an indication of their hopes for a world that is characterized by more justice, equity, and sustainability.

The results presented in this study emphasize the ongoing necessity of providing sustained assistance and enablement to young individuals in their endeavors to make meaningful contributions towards the establishment of sustainable value generation. It is imperative for governments, civil society organizations, educational institutions, and corporations to acknowledge the significance of youth engagement and furnish young people with the essential means, resources, and platforms to enhance their voices and initiatives. The results obtained in this study align with the findings of previous authors, such as Borojević and others [44], who have also highlighted the significance of certain challenges for the improvement in sustainability and the participation of young individuals in its achievement. These findings provide a justification for more research in this field of study.

Young individuals are considered to be crucial stakeholders and active participants in discussions related to SD and viability. They possess the entitlement to a sustainable future and have the right to be included in the process of change [44,76].

Greta Thunberg, a youth climate activist, has said: "It is our future on the line, and we must at least have a say in it. . . Being young is a great advantage, since we see the world from a new perspective, and we are not afraid to make radical changes. . . You must take action. You must do the impossible. Because giving up is never an option". Following her example, only with the complete understanding of all 17 SDGs and continuous efforts to do the impossible will humankind be a step closer to achieving SD.

**Author Contributions:** Conceptualization, T.B., M.M. and N.P.; Methodology, T.B.; Software, H.G.; Validation, M.R., N.M., J.A.R. and D.M.; Formal Analysis, T.B. and M.M.; Investigation, T.B.; Data Collection, all authors; Data Curation, H.G., M.R. and N.M.; Writing—Original Draft Preparation, T.B., N.P. and J.A.R.; Writing—Review and Editing, N.P. and M.M.; Visualization, J.A.R.; Supervision N.P. All authors have read and agreed to the published version of the manuscript.

**Funding:** This research received no external funding.

**Institutional Review Board Statement:** Not applicable.

**Informed Consent Statement:** Informed consent was obtained from all subjects involved in the study.

**Data Availability Statement:** The data presented in this study are available on request from the corresponding author.

**Acknowledgments:** The authors express their gratitude to the Editors and anonymous reviewers for their valuable feedback, which enabled the improvement and clarification of this paper.

**Conflicts of Interest:** The authors declare no conflict of interest.

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
