# Peer review of "Youth Participation for Sustainable Value Creation: The Role and Prioritization of SDGs"

_sustainability, doi:10.3390/su152316456_

Round 1
Reviewer 1 Report
Comments and Suggestions for Authors
I find this article difficult to read and understand.
The chapter on methodology should be reorganized, with precise information on the sample and procedures.
You should avoid phrases such as "Furthermore, it is not necessary for the distribution of random variables X1, X2,..., Xn to conform to the normality criterion in order to use parametric techniques"
The results should be presented in a consistent manner and with tables of interpretation assistance.
Discussions should be better articulated and supported by more robust and recent bibliography.
Reviewer 2 Report
Comments and Suggestions for Authors
A very interesting topic, indeed. In the abstract, please highlight the method of the analysis that you did in your study.
I strongly recommend dividing the first section between the introduction and the literature review. In the introduction, let us know the reason for dealing with this study and your objectives. The literature, that could present more sources in this topic should discuss similar frameworks focusing on the prioritization of SDGs approaches. You should mention the attributes of taking these specific regions into the sample. Did you take into account the structure of youth (percentage of total population, did you take into account education level, urban vs rural youth?). This is not related to how many respondents (their structure) but do the regions have some specifics to be chosen among others?
The methods of analysis are well explained.
What you wrote from page 7, it more belongs to the discussion section. So, you should give a discussion section. However, your result section is not present results as expected per the methodology. I see only descriptive statistics there.
Where are the results that you promised to show:
Non-parametric Mann-Withney U tests (when the variable of interest has two categories) and Kruskall Wallis H (when the variable of interest has three categories) were used to examine the relationships and the existence of certain legality between socio-demographic characteristics and attitudes in all analyzed countries. Non-parametric tests were used because respondents' answers were measured on a Likert scale (from 1 to 7).
To examine differences, we used independent t- and One-way ANOVA, while post-hoc comparisons were conducted using the Tukey Post Hoc Test. Differences between empirical and theoretical frequencies were tested by the Chi-Square Test. To examine individual predictive features of independent variables we used the Univariate Linear Regression. The Hierarchical Linear Regression determined a combination of predictors explaining the dependent variable in the highest percentage.
Without this, the result section is incomplete.
Round 2
Reviewer 1 Report
Comments and Suggestions for Authors
The authors have improved the manuscript
Reviewer 2 Report
Comments and Suggestions for Authors
The authors have made an additional effort to improve the paper. The structure has more clarity. The methodology section provides more precise clarification. I found the result section more explanatory.